# A U-Shaped Optical Fiber Temperature Sensor Coated with Electrospinning Polyvinyl Alcohol Nanofibers: Simulation and Experiment

**DOI:** 10.3390/polym14102110

**Published:** 2022-05-22

**Authors:** Yen-Lung Chou, Hsin-Yi Wen, Yu-Qiao Weng, Yi-Ching Liu, Chao-Wei Wu, Hsiang-Cheng Hsu, Chia-Chin Chiang

**Affiliations:** 1Department of Mechanical Engineering, National Kaohsiung University of Science and Technology, Kaohsiung 80778, Taiwan; 1106403103@nkust.edu.tw (Y.-L.C.); p467821@gmail.com (Y.-Q.W.); f108142140@nkust.edu.tw (Y.-C.L.); gn1204774@gmail.com (H.-C.H.); 2Department of Green Energy and Environmental Resources, Chang Jung Christian University, Tainan City 71101, Taiwan; hywen@mail.cjcu.edu.tw; 3Department of Chemical and Materials Engineering, National Kaohsiung University of Science and Technology, Kaohsiung 80778, Taiwan; 4Department of Aeronautical and Mechanical Engineering, Air Force Academy, Kaohsiung, No. Sisou 1, Jieshou W. Rd., Gangshan District, Kaohsiung City 82047, Taiwan; cafa95011@gmail.com

**Keywords:** bent optical fiber grating, electrospinning, polymer, temperature sensor

## Abstract

This study describes the fabrication of an electrospun, U-shaped optical fiber sensor for temperature measurements. The sensor is based on single mode fibers and was fabricated into a U-shaped optical fiber sensor through flame heating. This study applied electrospinning to coat PVA, a polymer, onto the sensor layer to reduce its sensitivity to humidity. The sensor is used to measure temperature variations ranging from 30 °C to 100 °C. The objectives of this study were to analyze the sensitivity variation of the sensor with different sensor layer thicknesses resulting from different electrospinning durations, as well as to simulate the wavelength signals generated at different electrospinning durations using COMSOL. The results revealed that the maximum wavelength sensitivity, transmission loss sensitivity, and linearity of the sensor were 25 dBm/°C, 70 pm/°C, and 0.956, respectively. Longer electrospinning durations resulted in thicker sensor layers and higher sensor sensitivity, that wavelength sensitivity of the sensor increased by 42%.

## 1. Introduction

Optical fiber sensors can detect a wide range of physical quantities by using optical signals as a means of transmission. Compared to typical electronic sensors, optical fiber sensors offer many advantages such as resistance to leakage, fire, radio frequency and other electromagnetic interference, and radiation. They are applicable over long distances, and useful in distributed sensing applications [1]. These properties lend particular advantages in many harsh environments, including current applications in industrial and defense contexts. In addition to taking physical quantity measurements, many scholars in recent years have fabricated various sensor layers by using nanomaterials [2], polymer materials [3], and magnetron sputtering [4]. These diverse sensor layers are used in humidity [5], biomedicine [6,7], electromagnetic field [8] detection as well as in fabricating optical fiber sensors with higher sensitivities.

The sensing principle of U-shaped optical fiber sensors is the whispering gallery mode (WGM) of optical fibers. WGM was introduced in 1910 by the English physicist Lord Rayleigh [9]. It was originally thought that this phenomenon occurs only in sound waves traveling along curved walls, but it was later discovered that similar effects occur in light waves as well. Tiegen Liu and their colleagues [10] pointed out that a special WGM wavelength spectrum is generated when the bend radius of an optical fiber is small enough that light propagation is enhanced due to the interference and coupling within the core and cladding layers in the bending region of an optical fiber. Multiple peaks and troughs can be observed in this special WGM wavelength spectrum. A team led by Xingling Peng [11] developed a U-shaped optical fiber-based temperature by coating the surface of the sensor with ink, bending it into a U shape, and then immersing it into an electroplating solution where it is electroplated from a temperature of 0 °C to 80 °C. As the temperature increases, the thermal stress created by the different coefficients of thermal expansion of the fiber and the coating altered the refractive index of the core and coating layers, thereby altering the bend radius and increasing the fiber’s bend loss.

Poly-(vinyl alcohol) (PVA) is an easily available polymer than consists of many hydroxyl groups in its molecular structure. It is widely applied in various domains as it is easily manufactured [12,13], biodegradable [14,15,16], and has excellent chemical resistance [17] and physical properties [18,19]. In addition, electrospinning has been adopted as a convenient technology to fabricate nanofibers in the microstructure [20,21], due to their large specific surface area and high porosity in the sensing area, high absorption ability, biocompatibility [22], and excellent performance in biosensing and environmental [23,24]. The conducted polymer nanofibers performed by the electrospun process have received considerable attention in the field of soft electronics owing to their promising conductivity, and transparency in utilizing flexible/stretchable electronic devices [25,26]. For example, Ning Chen and their colleagues proposed a highly sensitive humidity sensor with low-temperature sensitivity using a PVA-coated taper fiber [27]. The microscale diameter of the taper fiber allows it to be highly sensitive to its environmental medium. A research team led by Jia-Kai Wang proposed a double-D-shaped optical fiber sensor for temperature and humidity based on surface plasmon resonance (SPR) [28]. The two flat surfaces of the double-D were coated with ethanol as a temperature-sensitive material and PVA as a humidity-sensitive material, respectively. By applying thermogravimetric analysis (TGA), Seon Jeong Kim and their colleagues [13] found that PVA has an excellent thermal stability over the 30 to 100 °C range. Abolfazl Noorjahan and their colleagues [29] also found that PVA possesses excellent thermal expansion coefficients over the same 30–100 °C range. This study applies those findings to designing a U-shaped optical fiber sensor with electrospun PVA and experimentally analyzing how it is influenced by temperature changes.

## 2. Theory

As the propagation of light in an optical fiber is influenced by the surrounding physical conditions, the sensitivity of the fiber toward these physical conditions can be exploited as the working basis of an optical fiber sensor. Axial strain occurs on a fiber when it is stretched or compressed by an axially aligned force. Based on the theory of elasticity, Hooke’s law states that stress is directly proportional to strain when there are no shear forces, and horizontal deformation occurs along with longitudinal deformation. The strain created by an axial force can be expressed as follows:(1)$$\left(\begin{array}{c} e_{x}\\ e_{y}\\ e_{z} \end{array}\right)=\frac{F}{AY}\left(\begin{array}{ccc} 1 & -v & -v\\ -v & 1 & -v\\ -v & -v & 1 \end{array}\right)\left(\begin{array}{c} 0\\ 0\\ 1 \end{array}\right)=\frac{F}{AY}\left(\begin{array}{c} -v\\ -v\\ 1 \end{array}\right)$$
where $e_{x}$, $e_{y}$, and $e_{z}$ are the Cartesian strain elements; Y is Young’s modulus of the material; v is Poisson’s ratio of the material. F is the axial force and A is the fiber’s cross-section area. For silica dioxide, $Y=6.5\times 10^{10}\;N/m^{2}$ and v=0.17 [30,31]. Strain results in photoelasticity, that is, the refractive index increases proportionally with increasing strain. In a single-mode fiber (SMF) the propagation of light is essentially horizontal and its changes can be approximated as follows:(2)$$\Delta n_{eff}=-n^{3}\left[\left(1-v\right)p_{12}-vp_{11}\right]e_{z}/2=\gamma ne_{z}$$
where $\gamma =-n^{3}\left[\left(1-v\right)p_{12}-vp_{11}\right]/2$ is commonly known as the effective photoelastic coefficient. The photoelastic coefficients of bulk silica are $p_{11}=0.113$ and $p_{12}=0.252$ at a wavelength of 632.8 nm [32,33]. The horizontal tension applied to an optical fiber can be decomposed into axial tension and diagonal unidirectional tension. Since the cross-section of a compressed fiber is often longer than its horizontal length, substituting the derived strain into the photoelasticity equation yields:(3)$$\Delta \left(\frac{1}{n^{2}}\right)=\frac{\left(1+v\right)\left(1-2v\right)P}{Y}\left(\begin{array}{ccc} p_{11} & p_{12} & p_{12}\\ p_{12} & p_{11} & p_{12}\\ p_{12} & p_{12} & p_{11} \end{array}\right)\left(\begin{array}{c} 1\\ 1\\ -v \end{array}\right)$$

Changes in the refractive index can be expressed as:(4)$$\Delta n_{x}=\Delta n_{y}=-\frac{n_{0}^{3}}{2Y}\left(1+v\right)\left(1-2v\right)\left(p_{11}+p_{12}\right)P$$

The phase change in an optical fiber with a length *L* can be expressed as:(5)$$\Delta n_{\phi }=-\frac{n_{0}^{3}}{2Y}\left(1+v\right)\left(1-2v\right)\left(p_{11}+p_{12}\right)PkL$$

In U-shaped optical fiber sensors, a lower effective critical angle reduces the amount of light that can be propagated into the fiber, and the light beams are traced through several internal reflections. Multiple reflections inside a bent fiber with a constant bend radius do not increase the bending loss since the curvature is fixed when light beams are incident on the sidewall. Bending losses mainly happen where a fiber’s straight section transitions to a curved section (i.e., where the bend radius changes) [34]. Harris and their colleagues [35] studied this model and reported that total internal reflection occurs at the cladding layer of a U-shaped optical fiber sensor and subsequently creates the WGM. Hagen Renner and their colleagues further examined this issue and performed a pilot study on the transmission loss of a fiber. The authors reported that, when single-mode optical fibers experience bending, the transverse field distribution *ψ*(x,y) in the optical waveguides can be expressed as the two-dimensional scalar equation:(6)$$\nabla_{t}^{2}\psi \left(x,y\right)+\left[k^{2}n_{eff}^{2}\left(x,y\right)-\beta^{2}\right]\psi \left(x,y\right)$$
where k=2π/λ, λ and β represent the propagation constant, the wavelength of the leak mode, and the composite propagation constant, respectively. The effective refractive index distribution in the bent fiber can be expressed as follows [36]:(7)$$n_{eff}^{2}(x,y)=n^{2}\left(x,y\right)(1+2x/R)$$
where $n^{2}(x,y)$ represents the refractive index of the bent optical fiber, $n_{eff}^{2}(x,y)$ is the refractive index distribution of the bent optical fiber, R is the effective bend radius. When the fiber has an external coating and R is smaller than the critical bend radius ($R_{c}$), the equation can be shortened to:(8)$$R_{c}=2k^{2}n_{2}^{2}b/\gamma^{2}$$
where b is the fiber radius. When the peak-to-valley ratio of the bending transmission loss (η), bend radius variation (ΔR), and wave peak distance in a coated single model fiber are related, the wavelength variation (Δλ) can be expressed as follows:(9)$$\eta ≃{\left(\frac{n_{3}}{n_{2}}\right)}^{\frac{2}{3}}\left(1+\frac{R}{2b}\frac{n_{3}^{2}-n_{2}^{2}}{n_{3}^{2}}\right),R\ll R_{c}\;\mathrm{and}\;\gamma^{2}\ll \sigma^{2}$$
(10)$$\Delta R=\frac{3\lambda }{2n_{2}}{\left(\frac{R^{3}}{2b^{3}}\right)}^{\frac{1}{2}},R\ll R_{c}$$
(11)$$\Delta \lambda \approx \frac{3\lambda^{2}}{4n_{2}^{2}}{\left(\frac{R}{2b^{3}}\right)}^{\frac{1}{2}},\;R\ll R_{c}$$
where *λ* is the wavelength, *R* is the bend radius, *R_c_* is the critical bend radius. Based on this equation, the wavelength is affected by the parameters mentioned above when *R*
*≪ R_c_*, and the wavelength is affected by the aforementioned parameters. Therefore, varying the bend radius results in spectrum peak variation. The experiments in this study were based on the sensing mechanism of coated sensors.

## 3. Experimental

In this study, bent optical fibers were fabricated into U-shaped optical fibers, which were then fabricated into sensors by electrospinning temperature- and humidity-sensitive thread structures onto U-shaped curve. To make the U-shaped fiber foundation, a 30 mm mid-section was stripped from a single-mode fiber and secured inside a quartz glass tube, which was then placed onto a micro-positioning stage in order to set the optical fiber’s bend radius. The single-mode fiber was purchased from Corning SMF28 fiber at a 1310/1550 nm operating wavelength and the effective index of refraction was 1.4682 at a 1550 nm operating wavelength. The single-mode optical fiber was fabricated into a “U” shape using a flame heating process. A blowtorch was used to heat the loop uniformly until the required temperature is attained. Their two ends of the optical fiber are put into a hollow glass tube and fixed in the glass tube with UV resin, its packaging must be adequate to ensure that the diameter remains unchanged. The fiber was then allowed to cool, resulting in a U-shaped loop with a bend radius about 15 mm. Lastly, the loop was packaged in an application of UV-curable adhesive. The fabrication process is shown in Figure 1.

The temperature-sensing PVA layer was fabricated by electrospinning, using the following parameters: voltage: 18 kV; electrospinning solution: spinning polymer solution of 12.0 wt. % PVA; flow rate: 60 mm^3^/min; electrospinning distance: 200 mm; ambient temperature = 25 °C; electrospinning duration: 5, 10, and 15 min; relative humidity at 55% RH. PVA (MW 89,000–98,000, Sigma-Aldrich, Inc, St. Louis, MI, USA) fiber electrospinning was employed to coat both sides of the U-shaped optical fiber probe sensor, which was secured onto a collection plate covered with copper tape to complete the probe sensor. The experimental setup is shown in Figure 2. During electrospinning, the DC power supply provides either a positive and ground potentials. That positive electrode was connected to a hollow syringe while the ground potentials was connected to a metal screen. 18 kV DC voltage was applied to form a high-voltage electric field. The syringe was filled with electrospinning solution, and expressed at a fixed rate. The emerging PVA solution was then drawn out by the electric field into nanoscale-thickness threads to be collected on the fiber loop and metal screen. Electron microscope confocal images of PVA nanofiber coated sensor obtained were processed by using NIKON A1. The optical fiber was placed into a quartz glass tube and slightly adjusted and both ends of the fiber were fixed on a micro platform. The mold was secured onto a stage, and flame from a gas blowtorch was directed onto the loop. The diameter of the U-shape depends on the final bending radius on a micro platform to control each U-fibers. Therefore, the radius of each U-shaped fiber will be slightly different. Logically the more time the electrospinning process is on the thicker the layer that is deposited onto the collector. Figure 3 shows three individual probes, the radius of each U-shaped fiber is slightly different, so the radius of the U-shaped fiber in Figure 3 is not positively correlated with the length of time. Representative finished products are shown in Figure 3. Optical microscope (OM) images Figure 3a–c for times depended, and shows an optical microscope (OM) image Figure 3d for the bar U-shaped fiber, and SEM images Figure 3e,f are necessary to show the fiber morphology after the electrospun fibers process.

Afterwards, the sensor was evaluated at a range of temperatures using the setup in Figure 4. Fabricated sensors were (separately) connected to an amplified spontaneous emission (ASE) light source and an optical spectrum analyzer (OSA), and placed into an oven together with a k-type thermocouple. As the oven was heated from 30 °C to 100 °C, changes in the sensor’s optical properties were recorded by the OSA, while temperature control was monitored by the thermocouple. Measurements were recorded every 10 °C, and experiments were performed in three cycles.

## 4. Results and Discussion

### 4.1. Optical Simulation of the Electrospun PVA-Coated Sensor

This study utilized the Wave Optics module in COMSOL Multiphysics 6.0 (Build:318, COMSOL, Inc. Burlington, VT, USA, 2022) to simulate the spectrum generated by the electrospun PVA-coated sensor on the U-shaped optical fiber. The sensor radius, measured at different electrospinning durations, was 1724 μm (after electrospinning for 5 min), 1257 μm (after 10 min), and 1494 μm (after 15 min). Then, a 2D software simulation was constructed, as shown in Figure 5a, for mesh analysis (the complete mesh contains 923,492 region elements and 32,880 boundary elements, for an average element quality of 0.9301). The mesh analysis is shown in Figure 5b. Simulations and signal analyses were then performed to generate the spectrum and electric field analysis images.

### 4.2. Simulated Optical Spectra for the Electrospun PVA-Coated Sensors

The OSA was used to measure the spectrum generated by the electrospun PVA-coated sensor of the U-shaped optical fiber. The generated spectra are shown in Figure 6. The greatest transmission losses of the sensor were at 1523.6 nm (electrospinning duration of 5 min), 1496.8 nm (10 min), and 1450.6 nm (15 min). COMSOL was used to simulate the signal variation of different sensors, which was exported as the spectrograms shown in Figure 7, where it can be seen that the greatest transmission losses of the sensor were 1525 nm (simulated using the radius of the sensor electrospun for 5 min), 1500 nm (simulated using the radius of the 10-min sensor), and 1455 nm (simulated using the radius of the 15-min sensor). Comparing Figure 6 and Figure 7, it is found that the transmission loss peaks of the actual spectrum and the simulated spectrum are very close to each other, but has few errors and the wavelength errors are 1.4 nm (5 min of electrospinning time), 3.2 nm (10 min of electrospinning time) and 4.4 nm (15 min of electrospinning time), respectively. This is due to the error, in the bending radius of the measurement sensor caused by the thickness of the electrospun wire at different electrospinning times, resulting in the simulated spectrum and the actual spectrum not being identical. The longer the electrospinning time, the more the wavelength difference caused by the thickness increases proportionally, which is in accordance with Equation (11). Figure 8 shows the simulated energy field analysis results. It can be seen that, when light enters the bending region, total reflection is no longer maintained at the core–cladding interface, and a leakage mode forms. Reflections at the interface between the cladding layer and the external medium form a WGM, and the light rays are then coupled back into the core. The two different transmission pathways result in an interference pattern that aligns with Equation (6).

### 4.3. Temperature Experiments on the Electrospun PVA-Coated Sensor of the U-Shaped Optical Fiber

The U-shaped optical fiber sensors were tested for temperature stability in three cycles, with their spectral losses analyzed using an OSA. The results in Figure 9 show that the sensor electrospun for 5 min had a wavelength variation of 3.373 nm and a total transmission loss variation of 0.533 dB; the sensor electrospun for 10 min had a wavelength variation of 4.997 nm and a total transmission loss variation of 1.944 dB; the sensor electrospun for 15 min had a wavelength variation of 5.623 nm and a total transmission loss variation of 1.345 dB. The spectrograms clearly show that the signal peaks shift toward longer wavelengths at higher temperatures; this can be explained as due to the changes in the sensor radius from thermal expansion in the PVA (which has a coefficient of thermal expansion of 70 × 10^−6^/K). These findings are in line with the results derived from Equation (10), while the overall wavelength variation is in line with the results derived from Equation (11). The experimental results suggest that longer electrospinning durations result in thicker sensing layers and higher wavelength sensitivities, as high as 70 pm/°C. When the electrospinning time increases, the amount of PVA nanofilaments increases, and the wavelength sensitivity of the fiber also increases, which is due to the effect of the change in the radius of the U-shaped fiber. This can be determined by the transmission loss not sensitivity increases by electrospinning time increases. The greatest transmission losses in actual conditions were 1523.6 nm, 1496. 8nm, and 1450.6 nm, which were extremely similar to the simulation results. Figure 9d–f show that the average wavelength sensitivities at electrospinning durations of 5 to 15 min were 0.049 nm/°C, 0.059 dB/°C, and 0.079 nm/°C, respectively, and the mean transmission loss sensitivities were 0.008 dB/°C, 0.025 dB/°C, and 0.020 dB/°C, respectively. U-shaped optical fiber probe sensor was coated with PVA for measuring physical quantities, such as temperature and humidity. When the temperature was ample larger, the radius of the optical fiber was reduced to the point that the loop became a U shape that fits into the preset diameter. The curvature of the response is observed the linear behavior is not obeyed along with all of the temperature intervals, and linear behavior only in the initial temperature region. The measurements were compiled into the electrospinning duration–sensitivity relation graph shown in Figure 10. It can be seen that only the wavelength sensitivity increases linearly with the electrospinning duration, but not the transmission loss. The greatest increase in wavelength sensitivity was 42%, while the overall linearity of the sensor was 0.704 at maximum. These findings indicate the sensor’s excellent reliability under varying temperatures.

## 5. Conclusions

This sensor in varying temperatures via U-shaped optical fiber was performed through an electrospun nanofibers process. The unique structure and properties of electrospun nanofiber materials are provided through PVA polymer, and the spectrum simulates via COMSOL Multiphysics 6.0—Wave Optics module to perform electric field analysis. The findings indicate that sensitivity in resonance wavelength was increased as durations of electrospun nanofibers process increased caused by nanofiber properties of PVA and U-shaped optical fiber. The peak signal of the sensor has been shifted with the bending radii variation. When light inlets the bending region, total reflection is not consistent at the interface between the core and cladding layer, and a leakage mode forms in the bending region of a U-shaped optical fiber. Reflections occurring at the interface between the cladding and the external medium form a WGM. We then summarize the sensitivity in resonance wavelength and transmission loss of the sensor were 0.079 nm/°C and 0.025 dB/°C, respectively. The resonance wavelength sensitivity with durations of electrospun nanofibers process was increased, and the linearity of the sensor was 0.999.

## Figures and Tables

**Figure 1 polymers-14-02110-f001:**
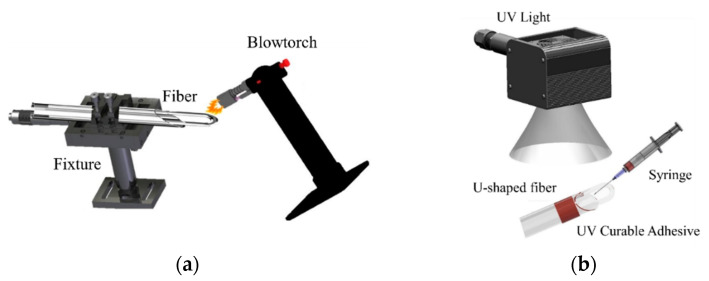
(**a**) Schematic of the fabrication of a U-shaped optical fiber via flame heating; (**b**) schematic of the UV curing and packaging process.

**Figure 2 polymers-14-02110-f002:**
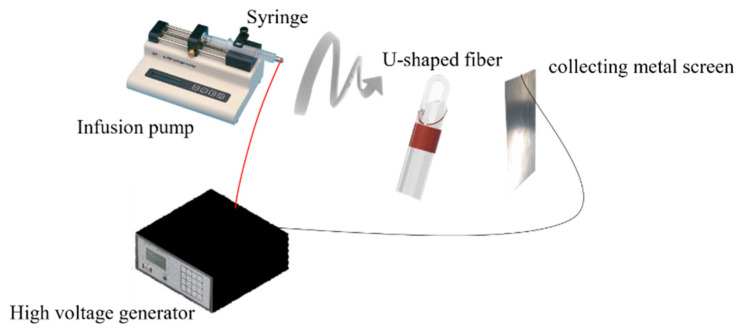
Schematic of electrospinning the PVA layer onto the sensor.

**Figure 3 polymers-14-02110-f003:**
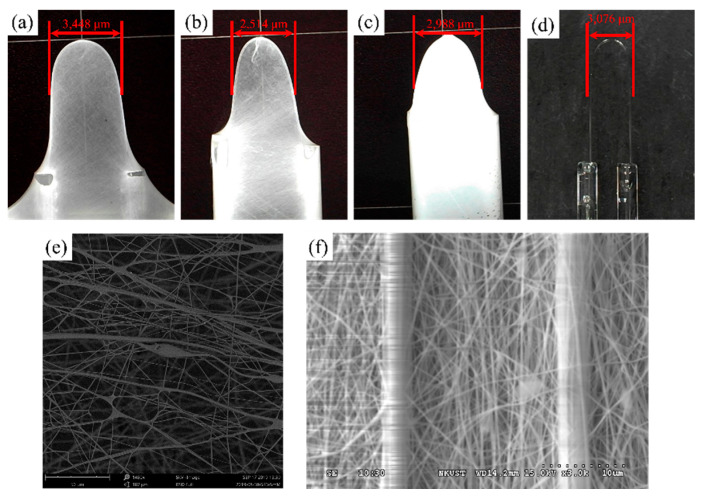
Electrospun U-shaped optical fiber products formed under different electrospinning durations (**a**) 5 min, diameter = 3448 μm; (**b**) 10 min, diameter = 2514 μm; (**c**) 15 min, diameter = 2988 μm; (**d**) optical microscope (OM) image for the bar U-shaped fiber; (**e**) SEM im-age for electrospun fibers; (**f**) SEM image to show the fiber morphology after the electrospun PVA fibers.

**Figure 4 polymers-14-02110-f004:**
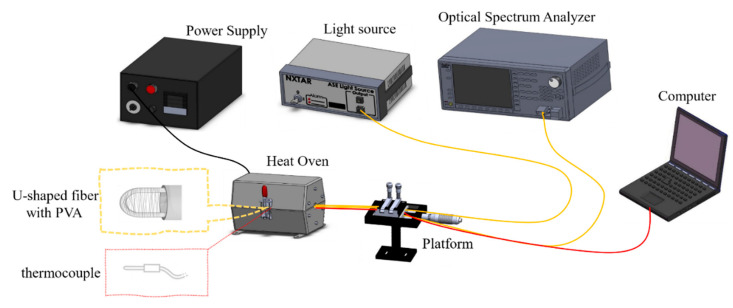
Setup for the temperature experiments. The electrospun PVA-coated sensor of the U-shaped optical fiber is connected to an ASE and an OSA and was placed into a heat oven with a k-type thermocouple.

**Figure 5 polymers-14-02110-f005:**
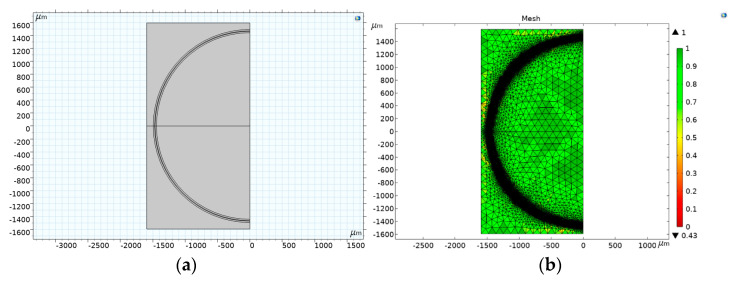
(**a**) Optical fiber model and (**b**) mesh analysis.

**Figure 6 polymers-14-02110-f006:**
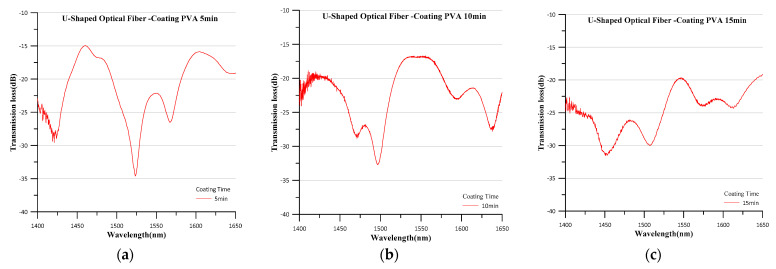
Spectra generated by the electrospun PVA-coated sensors with electrospinning durations of (**a**) 5 min, (**b**) 10 min, and (**c**) 15 min.

**Figure 7 polymers-14-02110-f007:**
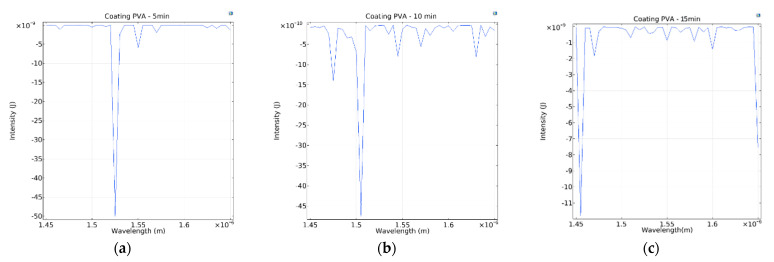
Sensor spectrum simulated in COMSOL for sensors with electrospinning durations of (**a**) 5 min, (**b**) 10 min, and (**c**) 15 min.

**Figure 8 polymers-14-02110-f008:**
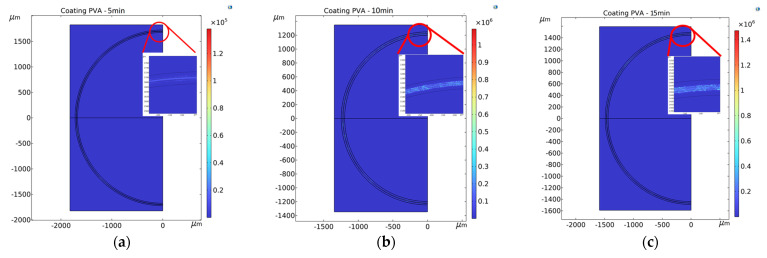
Energy fields simulated in COMSOL for sensors with electrospinning durations of (**a**) 5 min, (**b**) 10 min, and (**c**) 15 mined in the main text as Figure 1 etc.

**Figure 9 polymers-14-02110-f009:**
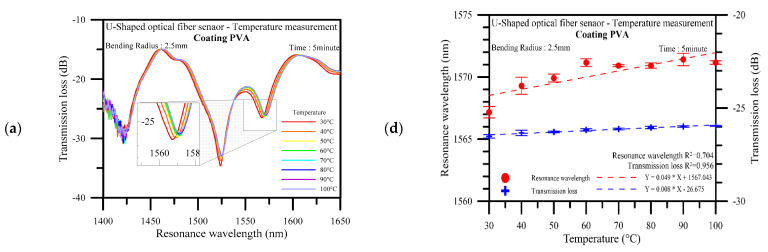
Spectrum generated at an electrospinning duration of (**a**) 5 min; (**b**) 10 min; and (**c**) 15 min; spectrum analysis with the error bars at an electrospinning duration of (**d**) 5 min; (**e**) 10 min; and (**f**) 15 min.

**Figure 10 polymers-14-02110-f010:**
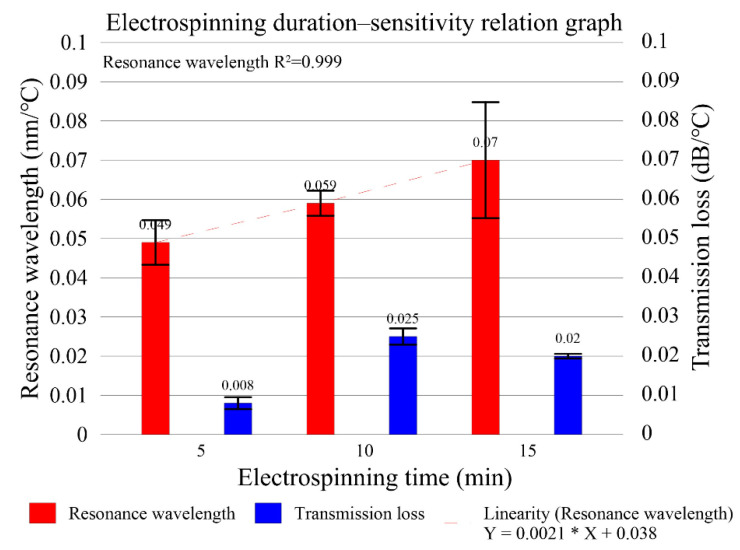
Electrospinning duration–sensitivity relation graph.

## Data Availability

Not applicable.

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
