# Peer review of "A U-Shaped Optical Fiber Temperature Sensor Coated with Electrospinning Polyvinyl Alcohol Nanofibers: Simulation and Experiment"

_polymers, 2022, doi:10.3390/polym14102110_

Round 1

Reviewer 1 Report

The study describes the fabrication of an electrospun, U-shaped optical fiber sensor for temperature measurements. The impact of variations in the electrospinning duration was investigated and the maximum wavelength sensitivity, transmission loss sensitivity, and linearity of the sensors were determined. The work is well performed and clearly explained, and appropriate for the Polymers journal, so I recommend its publication. Some slight observations are:

  1. Along with the manuscript, many quantities seem to be not alighted with respect to the baseline of the sentences (they are above the line). Maybe is a problem related to the creation of the pdf file.
  2. In Figure 6 section 4.2 a chemical explanation for the observed differences in profiles is missing. It is clear that the electrospinning duration has a great impact on the signals, could the authors propose an explanation according to the changes occurring in the evolution of the electrospinning procedure? Apparently, in section 4.3 some proposal is given (The experimental results suggest that longer electrospinning durations result in thicker sensing layers, lines 230-231), could this statement be verified in some form?
  • In Figure 9, although an increase in the resonance wavelength with temperature is observed, some curvature of the response is observed, and probably the linear behavior is not obeyed along with all the temperature intervals. Cold the authors get deeply into the explanation of this behavior and its practical implications?

Author Response

Dear reviewer,

I am writing to you regarding our paper entitled” An U-shaped optical fiber temperature sensor coated with electrospinning polyvinyl alcohol nanofibers: simulation and experiment”, Manuscript Number: polymers-1732734.

We greatly appreciate your comments and those of reviewers. The technical comments have helped us to improve our manuscript considerably. The following is a list of the original comments along with our responses, which also specify how and where the manuscript was modified. Changes made in the manuscript are marked in a red font and using the “Track Changes” function. The revisions to the manuscript were made in consultation with all of the coauthors, and each author has given their approval to the final form of the revised manuscript revision.

Sincerely,

Chia-Chin Chiang

Professor, National Kaohsiung University of Science and Technology

TEL: +886-7-3814526 ext 15340

FAX: +886-7-3831373

Postal address: No. 415, Jiangong Rd., Sanmin Dist., Kaohsiung City 80778, Taiwan

Reviewer 2 Report

The authors explain the manufacture process and performance of a U-shaped sensor containing PVA electrospun fiber mat. The collection time, thus the thickness of the fibermat,   during the electrospinning process is used as the parameter. The authors performed optical spectrum analysis (OSA) at different temperatures, to prove the capabilities of the fabricated U-shaped sensor. The COMSOL software was used to simulate the OSA spectra.

The work should be considered to be published after the following points are addressed:

line 36: What are “high polymer materials”?

The introduction is short and to the point. However, it would be great interest to the reader if the authors could include schematics as of how the U-shaped sensor works and its structure. Also, there are only 22 references, out of which only 3 references from 2020 and onward. the authors should update the article with newer studies.

The authors should structure the article as Introduction, Experimental (methods), Results and discussion, Conclusion and so on…

line 136. What is the material the is used to produce the single-mode fiber, that is bent into a U shape?

Eq 1: please also define F and A, for clarity

line 152: please provide the polymer solution concentration (and why you choose that particular concertation) and the relative humidity, while electrospinning.

Figure 1 should be more informative and better adhere to the description in the text, meaning if the reader is not familiar with the process, the figure does not convey sufficient information to understand the process, while the purpose of the figure 1 should just that.

When dealing with electrospun fibers SEM images are necessary to show the fiber morphology and diameter. The average fiber diameter also should be reported to see if they are the same for all the cases or not.

line 150: The DC power supply provides either a positive and ground potentials or negative and ground potentials, not positive and negative

Figure 3. Logically the more time the electrospinning process is on the thicker the layer that is deposited onto the collector; however it is not the case here. Please explain. Also please specify where did you measure the diameter, a certain distance from the tip? What kind of microscopy do you use? Also please provide an image before the electrospinning process of the U shaped sensors.

line 172-174 is not necessary

line 195: “Comparing Figures 6 and 7, the actual and simulated wavelengths for 195
peak transmission loss were found to be extremely similar.”

This is an overstatement, and the authors should spend more time on explaining the differences in the figures. The effect of electrospinning time and resulted OSA spectra should be also discussed in more detail.

Figure 9. d, e and f should show error bars, since the experiments were replicated three times. Without the error bars, the data could be misleading.

Figure 10 Show error bars and trend line R2 value please

The conclusion section should augment the abstract and conclude the findings, state the major points of the study and state the correlations. The explanation of the phenomena belongs to the results and discussion section. Please rewrite the conclusion to project better the major findings of the work.

I don’t think using references 14 and 22 is adequate in the conclusion section.

Author Response

(The authors gave the same response as above.)
